# Chromatin determinants impart camptothecin sensitivity

Fabio Puddu[1,*] (ID), Israel Salguero[1], Mareike Herzog[1,2], Nicola J Geisler[1], Vincenzo Costanzo[3] &
Stephen P Jackson[1,**] (ID)

## Abstract

Camptothecin-induced locking of topoisomerase 1 on DNA generates a physical barrier to replication fork progression and creates topological stress. By allowing replisome rotation, absence of the Tof1/Csm3 complex promotes the conversion of impending topological stress to DNA catenation and causes camptothecin hypersensitivity. Through synthetic viability screening, we discovered that histone H4 K16 deacetylation drives the sensitivity of yeast cells to camptothecin and that inactivation of this pathway by mutating H4 K16 or the genes *SIR1-4* suppresses much of the hypersensitivity of *tof1*Δ strains towards this agent. We show that disruption of rDNA or telomeric silencing does not mediate camptothecin resistance but that disruption of Sir1-dependent chromatin domains is sufficient to suppress camptothecin sensitivity in wild-type and *tof1*Δ cells. We suggest that topoisomerase 1 inhibition in proximity of these domains causes topological stress that leads to DNA hypercatenation, especially in the absence of the Tof1/Csm3 complex. Finally, we provide evidence of the evolutionary conservation of this mechanism.

**Keywords** camptothecin; H4-K16; SIR complex; synthetic viability; Tof1
**Subject Categories** Chromatin, Epigenetics, Genomics & Functional Genomics; DNA Replication, Repair & Recombination

## Introduction

Separation of the two parental DNA strands during DNA replication creates positive supercoiling ahead of the replication fork. Such over-winding hinders replisome progression and must be removed for DNA replication to be completed. In eukaryotes, two DNA topoisomerases, Top1 and Top2, cooperate to allow DNA replication and segregation. The main DNA topoisomerase that relaxes positive supercoiling during DNA replication is considered to be Top1, a type-IB topoisomerase, while Top2 activity seems to be concentrated behind replication forks [1–3]. Despite the importance of DNA uncoiling for replication, *Saccharomyces cerevisiae* cells lacking Top1 can fully replicate their genome because in the absence of Top1, positive supercoils can either be relaxed directly by Top2 [4,5] or indirectly by rotation of replication forks along their axes, converting impending positive supercoiling into intertwines/catenation between the two daughter DNA strands [6]. The catenation generated in this way is an obstacle to chromosome segregation and must be resolved by Top2, a type II topoisomerase, before the onset of mitosis [3,7]. In contrast to Top1, Top2 is essential in yeast cells because a certain amount of catenation is generated even in wild-type cells, possibly because Top1 cannot relieve topological stress between replisomes converging towards replication termination zones [8]. Consistent with this model, increased fork rotation has been observed when replication forks approach stable fork-pausing structures, such as centromeres, tRNA genes, inactive replication origins [9], and potentially retrotransposon long terminal repeats (LTRs) and transcriptionally repressed chromatin [10,11].

To reduce the requirement for decatenation, replisome rotation is normally restricted by the Tof1/Csm3 complex [9], the yeast homolog of the mammalian Timeless/Tipin complex. Tof1 and Csm3 are also crucial for proper pausing of replication forks at replication fork barriers present in the tandem arrays that form the large ribosomal DNA (rDNA) locus [12]. Independently of these functions, the Tof1/Csm3 complex also interacts with Mrc1 [13], which functions as an adaptor to transmit signals from the apical replication-checkpoint kinase Mec1 to the transducer kinase Rad53 during replication stress induced by nucleotide depletion [14]. The fact that *tof1*Δ strains, similar to *mrc1*Δ strains, show synergistic phenotypes in combination with loss of Rad9—the other major checkpoint adaptor protein in *S. cerevisiae*—suggests that the Tof1/Csm3 complex recruits Mrc1 for the purpose of Rad53 activation [12,15]. In this regard, it is noteworthy that Mrc1 also has checkpoint-independent functions and can be recruited to replication forks independently of Tof1/Csm3 [14,16,17].

Despite the above findings, certain results have remained unexplained, and the precise roles of the Tof1/Csm3 complex have remained elusive. For instance, *tof1*Δ and *csm3*Δ yeast strains were shown to be hypersensitive to high doses of camptothecin [18], a drug that induces DNA double-strand DNA breaks (DSBs) during S

1 The Gurdon Institute and Department of Biochemistry, University of Cambridge, Cambridge, UK
2 The Wellcome Trust Sanger Institute, Hinxton, Cambridge, UK
3 IFOM (Fondazione Istituto FIRC di Oncologia Molecolare), Milan, Italy
  *Corresponding author. Tel: +44 1223 334088; E-mail: f.puddu@gurdon.cam.ac.uk
  **Corresponding author. Tel: +44 1223 334088; E-mail: s.jackson@gurdon.cam.ac.uk

phase by trapping Top1 in a covalent complex with DNA. These strains, however, are not hypersensitive to other agents that induce DSBs, such as ionising radiation, or to drugs such as hydroxyurea that affect S phase progression [18], suggesting that the camptothecin hypersensitivity of *tof1Δ* and *csm3Δ* strains might arise through topologically stressed DNA structures generated by Top1 inhibition rather than from DNA damage per se [19,20].

Here, we show that histone H4 K16 deacetylation by the yeast sirtuin complex drives the sensitivity of wild-type cells to camptothecin. Our results also show that the disruption of chromatin domains bearing deacetylated H4 K16 rescues the camptothecin hypersensitivity of *tof1Δ* and *csm3Δ* cells, suggesting that the increased sister chromatid catenation generated in the absence of these proteins promotes camptothecin toxicity. Finally, we show that the role of sirtuins in driving camptothecin sensitivity in *S. cerevisiae* is evolutionarily conserved in the yeast *Schizosaccharomyces pombe* and in human cells.

## Results

To better understand the roles of the Tof1/Csm3 complex during DNA replication, we investigated the basis for the camptothecin hypersensitivity of *TOF1*- or *CSM3*-deleted cells. This hypersensitivity arises from the well-established trapping of Top1 in a covalent complex with DNA, as shown by the fact that it was rescued by *TOP1* deletion (Fig 1A). Notably, *mrc1Δ* strains were not hypersensitive to camptothecin (Fig 1A; [18]), indicating that a defect in replication-checkpoint activation does not explain the camptothecin hypersensitivity of *tof1Δ* or *csm3Δ* strains. Moreover, this hypersensitivity does not appear to arise from issues connected to fork pausing at the replication fork barrier on rDNA, as pausing-deficient *fob1Δ* strains were not hypersensitive to camptothecin, and *FOB1* deletion did not alleviate the camptothecin hypersensitivity of a *csm3Δ* strain (Fig 1B).

### SIR gene mutations suppress camptothecin hypersensitivity of *tof1Δ*/*csm3Δ* cells

To understand the origin of the hypersensitivity of *tof1Δ* and *csm3Δ* strains to camptothecin, we carried out a synthetic viability genomic screening [21] to identify mutations capable of suppressing such hypersensitivity (Fig 1C). We plated approximately $1 \times 10^7$ cells on a YPD plate supplemented with 20 μM camptothecin (Fig EV1A), isolated sixteen resistant colonies, and verified that they indeed displayed both resistance to camptothecin and to the antibiotic G418, a readout for *TOF1* deletion, which was later confirmed by whole-genome sequencing (Figs 1D and EV1B). This validation ensured that the cells isolated did not merely survive camptothecin treatment, but carried genetic (or epigenetic) marks conferring camptothecin resistance. We then sequenced their genomic DNAs to identify candidate mutations responsible for the suppression phenotype (all the mutations identified in each strain are listed in Table EV1). Two of the sixteen strains—the most resistant ones—carried mutations that inactivated *TOP1*, which encodes the drug target. Three strains carried either of two nonsense mutations that inactivated *SIR3*, while eight of the remaining strains carried a nonsense mutation inactivating *SIR4* (Fig 1D; premature stop codons are designated

by a Δ following the position of the last amino acid residue encoded by the truncated gene). Importantly, we validated these putative drivers of camptothecin resistance by directly introducing deletions of *SIR3* and *SIR4* in *tof1Δ* and *csm3Δ* strains and establishing that *SIR3* or *SIR4* inactivation suppressed camptothecin hypersensitivity (Fig 2A). In the three remaining suppressor strains—the weakest suppressors—we could not identify any mutation responsible for the suppression. In one of these, no mutations were detected, while the other two carried point mutations in *IME2* (inducer of MEiosis, which is not expressed in exponentially growing cells) or *IRC15*. However, ensuing studies established that neither *IME2* nor *IRC15* deletion suppressed the camptothecin hypersensitivity of *tof1Δ* cells (Fig EV1C and D; the reasons for the decreased camptothecin sensitivity of these three strains therefore remain to be defined).

Sir3 and Sir4 form a ternary protein complex with the histone deacetylase catalytic subunit Sir2 (reviewed in [22]), with removal of any of the three subunits inactivating the transcriptional silencing functions of the complex [23]. Significantly, we established that loss of Sir2 alleviated the camptothecin hypersensitivity of *tof1Δ* cells to a similar extent as conferred by Sir3 or Sir4 loss (Figs 2B and EV2B, lower panel). Furthermore, by increasing the concentration of camptothecin, we found that deletion of *SIR2, SIR3*, and *SIR4* also promoted camptothecin resistance in a wild-type yeast background (Figs 2B and EV2B, upper panel). Interestingly, in wild-type cells, deletion of *SIR2* suppressed camptothecin sensitivity to a similar extent as conferred by *TOP1* deletion, and combining the deletions did not further increase resistance to camptothecin (Fig EV1E). By contrast, *SIR2* deletion did not alleviate the strong camptothecin hypersensitivity of a *rad51Δ* strain, which is severely defective in repairing DSBs induced by camptothecin (Fig 2C). These data indicated that the SIR complex is a major mediator of camptothecin sensitivity, but crucially, inactivation of the SIR complex does not act as a general suppressor of camptothecin toxicity, for example by reducing Top1 activity, cell permeability to camptothecin, or DSB induction by camptothecin.

### SIR proteins mediate camptothecin sensitivity via histone H4-K16 deacetylation

To assess whether loss of the deacetylase activity of the Sir complex was responsible for the suppression of *tof1Δ* hypersensitivity to camptothecin, we used the small-molecule Sir2 inhibitor, sirtinol [24]. This work established that addition of 20 μM sirtinol strongly suppressed the camptothecin sensitivity of a *tof1Δ* strain (Fig 2D). While Sir2 homologs in higher eukaryotes have been implicated in deacetylating proteins involved in DNA repair, such as PARP1, Ku70, and CtIP [25–27], the prime target for *S. cerevisiae* Sir2 is histone H4 lysine 16 (H4-K16), which is found in an acetylated state through much of the transcriptionally active yeast genome. In *S. cerevisiae*, deacetylation of this residue by Sir2 allows binding of Sir3, thus recruiting further Sir2 that removes acetylation marks from flanking H4-K16 residues, a process that is then propagated to produce a transcriptionally silent heterochromatic state [22]. To explore whether the relevant target for Sir2 in relation to its effects on the camptothecin sensitivity of *tof1Δ* cells was H4-K16, we mutated this residue to glutamine (Q), a residue that mimics a constitutively acetylated lysine and abrogates Sir3 binding [28]. Strikingly, this *hhf-K16Q* mutation suppressed the camptothecin

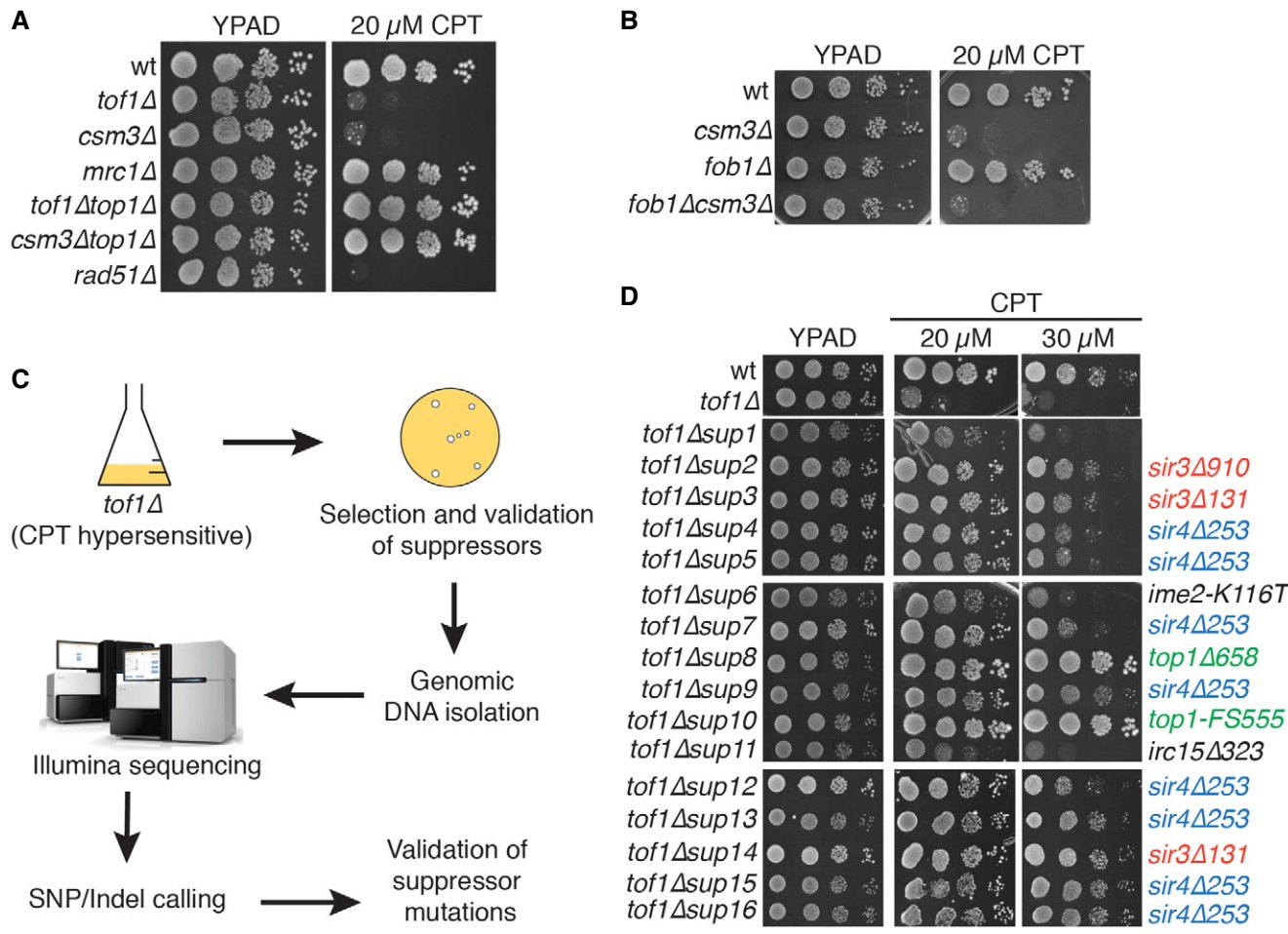

**Figure 1. A synthetic viability screening to identify the cause for the hypersensitivity of *tof1Δ* yeast cells to camptothecin.**

A  Loss of Tof1 and Csm3 but not Mrc1 causes hypersensitivity to camptothecin in a Top1-dependent manner.
B  Loss of pausing at the replication fork barrier on rDNA does not affect camptothecin hypersensitivity.
C  Outline of the procedure for a synthetic viability screen.
D  Synthetic viability screening identifies *sir3* and *sir4* alleles as suppressors of the camptothecin hypersensitivity of *tof1Δ* strains.

hypersensitivity of a *tof1Δ* strain, and at higher doses also reduced the camptothecin sensitivity of a wild-type strain (Fig 2E). Similarly, mutation of H4-K16 to glycine (G), which prevents binding by Sir3 [28], strongly counteracted the camptothecin sensitivity of both *tof1Δ* and wild-type cells. Taken together, these results highlighted a correlation between chromatin association of the SIR complex and camptothecin sensitivity.

### A deacetylated H4-K16 template promotes camptothecin-induced mitotic arrest

To further explore how *TOF1* or *CSM3* deletion causes camptothecin hypersensitivity, we took advantage of the fact that camptothecin treatment of synchronised wild-type cells released from G1 into S phase leads to a prolonged G2/M cell cycle delay [29]. We first assessed the effect of *TOF1* and *CSM3* deletion on this particular phenotype by arresting wild-type, *tof1Δ* and *csm3Δ* cultures in G1 by alpha-factor treatment, and then releasing them from this arrest

either in the presence or in the absence of camptothecin. As expected, wild-type cells treated in this way with camptothecin did not delay bulk DNA replication compared to strains released in the absence of camptothecin, although they did exhibit delayed exit from the subsequent mitosis (Fig 3A). Significantly, compared to wild-type controls, cells deleted for *TOF1* or *CSM3* arrested for longer periods of time in G2/M following camptothecin treatment (Fig 3A bottom panels), a phenotype that correlated with persistence of the mitotic cyclin, Clb2 (Fig 3B). Nevertheless, these cells eventually re-entered the cell cycle and continued proliferating, consistent with the fact that *tof1Δ* and *csm3Δ* strains were not killed by acute camptothecin treatment (Fig 3C; note that a repair-defective *rad51Δ* strain was hypersensitive even to acute camptothecin treatment).

Collectively, the data we had obtained supported a model in which the mechanism by which the SIR complex yields camptothecin sensitivity is via effects on H4-K16 deacetylation. In this regard, we reasoned that the SIR complex might impart

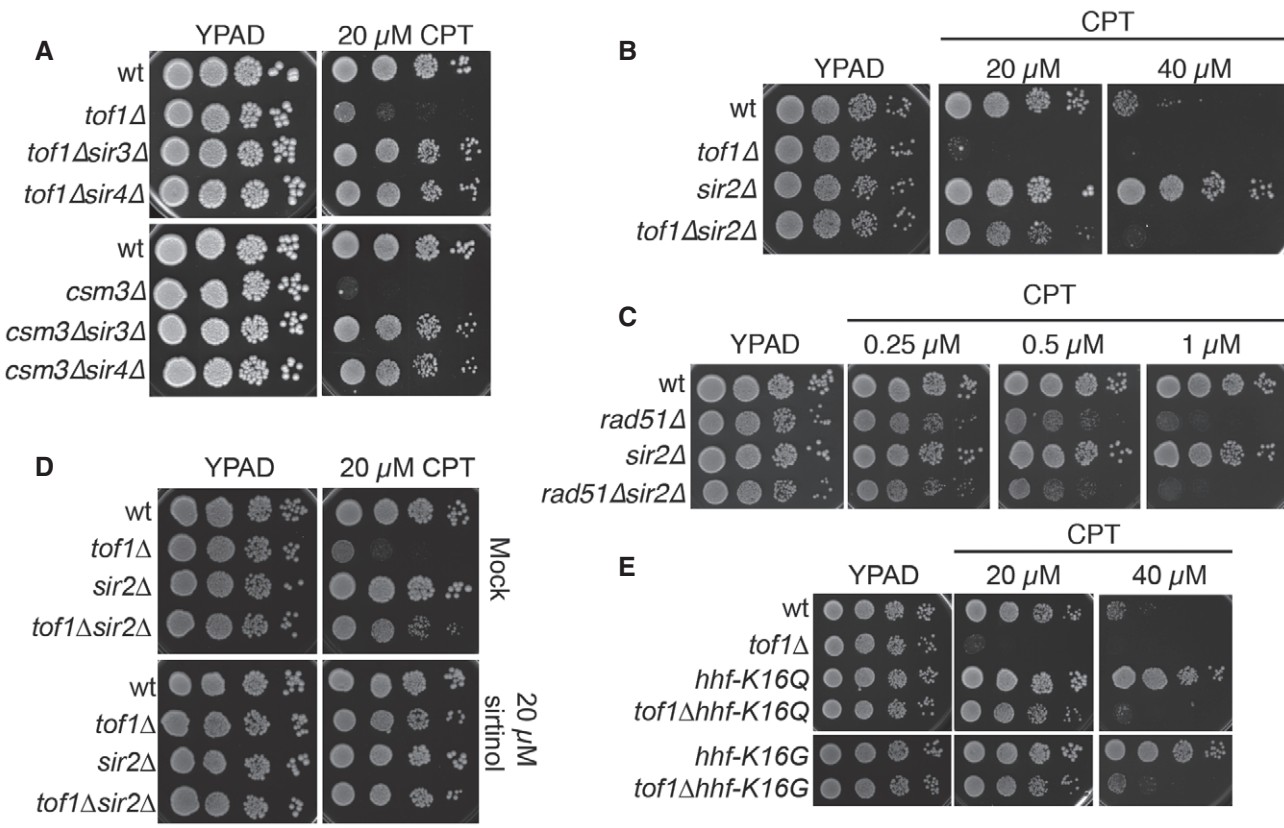

**Figure 2.  Loss of the SIR complex suppresses camptothecin hypersensitivity of *tof1Δ* yeast strains.**

A   Deletion of *SIR3* or *SIR4* suppresses the camptothecin hypersensitivity of *tof1Δ* and *csm3Δ* cells.

B   Deletion of *SIR2* also suppresses camptothecin hypersensitivity of *tof1Δ* cells and reduces camptothecin sensitivity of a wild-type strain.

C   Deletion of *SIR2* cannot suppress camptothecin hypersensitivity of a *rad51Δ* strain.

D   Inhibition of Sir2 deacetylase activity with sirtinol suppresses camptothecin hypersensitivity of *tof1Δ* cells.

E   Mutations that mimic a permanently acetylated H4-K16 (K16Q) or that remove the binding site for Sir3 (K16G) also suppress camptothecin sensitivity of wild-type and *tof1Δ* strains.

camptothecin sensitivity either by deacetylating newly incorporated histone H4 during DNA replication, or by it broadly promoting a condensed chromatin template that impairs DNA replication in the presence of camptothecin. If Sir2 deacetylation activity during S phase promoted camptothecin sensitivity, one would expect that addition of sirtinol after the release from G1 would circumvent the extended mitotic delay induced by camptothecin in *tof1Δ* cells. Conversely, if broad acetylation of the chromatin template was required to rescue the *tof1Δ* phenotype, sirtinol should lead to suppression of extended mitotic delay only if *tof1Δ* cells were pre-grown in the presence of sirtinol. To discriminate between these two hypotheses, we grew *hml* and *hml, tof1Δ* cells either in the presence or in the absence of sirtinol, and then synchronised them in G1 by addition of alpha-factor (Fig 3D). We used a mutant *hml* background because sirtinol makes wild-type cells insensitive to alpha-factor by derepressing the *HML/R* (*HM*) loci [24] (importantly, as shown in Fig EV1F, *HML* mutation did not affect camptothecin sensitivity). We then released the G1-synchronised cells into S phase in the presence of camptothecin alone, or in the presence of camptothecin plus sirtinol. While addition of sirtinol after the G1 release was not sufficient to rescue the mitotic delay of

*tof1Δ* cells (Figs 3D and EV1G), pre-growing *tof1Δ* cells in the presence of sirtinol suppressed their mitotic delay, whether or not sirtinol was present during the subsequent camptothecin treatment. A similar effect, albeit smaller, could be observed in a wild-type strain (Figs 3D and EV1G). Taken together, these findings supported a model in which camptothecin leads to replication-associated problems that arise within chromatin regions containing deacetylated H4-K16, with cells lacking Tof1 or Csm3 being particularly sensitive to such problems.

## Multiple HM-like chromatin regions govern camptothecin sensitivity

The yeast genome contains three well-studied heterochromatic regions that are transcriptionally silenced by SIR proteins: the rDNA array, subtelomeric regions and the cryptic mating-type loci (Fig 4A–C). To establish whether loss of rDNA silencing mediated the suppression of *tof1Δ* camptothecin hypersensitivity upon SIR protein loss, we used a strain carrying a deletion of the entire rDNA locus complemented by a multi-copy plasmid containing the rDNA repeat unit [30]. We found that deletion of the rDNA locus did not

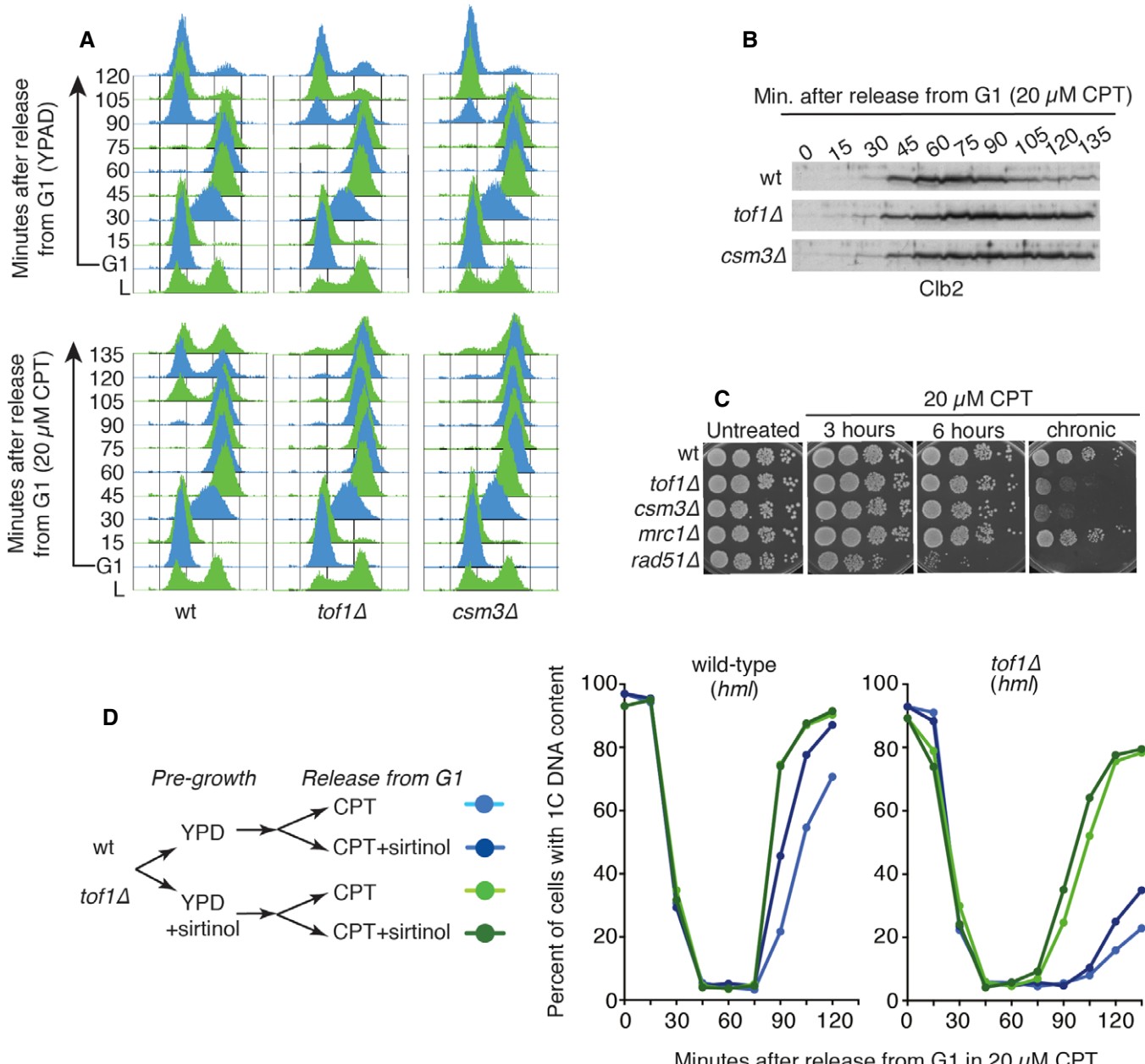

**Figure 3. A deacetylated H4-K16 template mediates sensitivity to camptothecin during DNA replication.**

A  A wild-type yeast strain released into S phase in the presence of 20 μM camptothecin does not exhibit delayed progression through S phase, but exhibits delayed progression through the subsequent mitosis. In the absence of Tof1 or Csm3, camptothecin-treated cells remain arrested in G2/M for longer periods of time than wild-type cells.

B  *tof1Δ* and *csm3Δ* cells released into S phase in the presence of camptothecin exhibit delayed destruction of the mitotic cyclin Clb2.

C  *tof1Δ* and *csm3Δ* cells are not hypersensitive to transient camptothecin treatment.

D  *tof1Δ* cells and congenic wild-type cells were pre-grown either in the absence or in the presence of sirtinol. They were subsequently synchronised in G1 and released into S phase in the presence of camptothecin, either with or without sirtinol. Cell cycle progression was monitored by FACS analysis. Quantification of G1 cells shows that sirtinol addition during camptothecin treatment does not suppress the mitotic delay of *tof1Δ* cells, while pre-growth in the presence of sirtinol is sufficient to suppress the camptothecin hypersensitivity phenotype of *tof1Δ* cells. A representative experiment is shown.

Source data are available online for this figure.

reduce the hypersensitivity of *tof1Δ* cells to camptothecin (Fig 4A), indicating that this genomic region is not the prime target of the SIR complex that mediates camptothecin toxicity in *tof1Δ* cells. This

notion was also supported by the fact that, while we observed suppression of camptothecin sensitivity with *sir2Δ*, *sir3Δ*, or *sir4Δ*, silencing of the rDNA locus only requires Sir2, with *SIR4* deletion

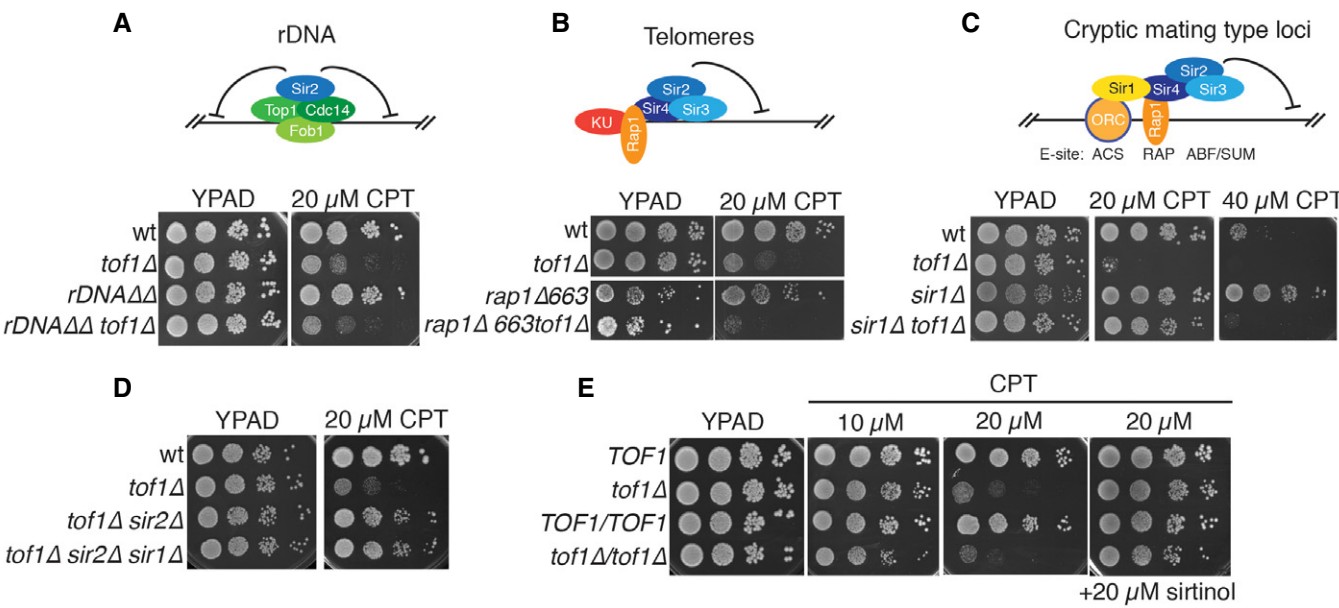

**Figure 4.  Disruption of Sir1-dependent silencing suppresses camptothecin hypersensitivity of *tof1Δ* cells.**

A  Disruption of the rDNA locus is not sufficient to suppress camptothecin hypersensitivity of *tof1Δ* cells.
B  A mutation in *RAP1* that disrupts telomeric silencing does not suppress camptothecin hypersensitivity of *tof1Δ* cells.
C  Deletion of *SIR1* suppresses camptothecin sensitivity in wild-type and *tof1Δ* cells.
D  *SIR1* deletion is epistatic with *SIR2* deletion with respect to suppression of camptothecin hypersensitivity of *tof1Δ* strains.
E  Homozygous *tof1Δ/tof1Δ* diploid cells are as sensitive to camptothecin as *tof1Δ* haploids, and their hypersensitivity can be alleviated by sirtinol.

actually increasing rDNA silencing by delocalising Sir2 from telomeres [31].

To determine if loss of subtelomeric silencing could rescue the camptothecin hypersensitivity of *tof1Δ* cells, we employed a strain carrying a C-terminal truncation of Rap1 (*rap1Δ663*), the so-called *rap1-17* allele. This mutation completely disrupts transcriptional silencing at telomeres (telomere position effect) and partially affects silencing of the cryptic mating-type locus *HML* [32]. While strains carrying the *rap1Δ663* allele grew slower than wild-type strains, presumably due to the role of Rap1 in regulating transcription of genes involved in ribosome formation and glycolysis [33,34], they did not display altered sensitivity to camptothecin (Fig 4B). The *rap1Δ663* mutation also failed to suppress the camptothecin hypersensitivity of *tof1Δ* cells (Fig 4B), indicating that loss of telomere position effect does not promote survival in the presence of this drug.

At the cryptic mating-type loci *HML* and *HMR*, silencing is established by replication origin recognition complex (ORC)-mediated recruitment of Sir1, which then attracts the SIR complex via an interaction with Sir4 [35,36]. Sir4 binding is also stabilised by an interaction with Rap1, which binds to its DNA consensus sequence located next to the ORC binding site ACS (ARS consensus sequence, Fig 4C). For these reasons, deletion of *SIR1* results in partial loss of silencing at the cryptic mating-type loci, but does not affect telomeric or rDNA silencing [37]. Strikingly, we found that *SIR1* deletion strongly alleviated the camptothecin hypersensitivity of a *tof1Δ* strain (Fig 4C). Importantly, *SIR1* deletion did not further improve the survival of *tof1Δsir2Δ* strains, suggesting that Sir1 mediates camptothecin sensitivity entirely via its connection to the Sir2/3/4 complex (Fig 4D). Furthermore, similar to what we had observed

for *SIR2*, *SIR3*, or *SIR4* deletion, disruption of *SIR1* also decreased the sensitivity of a wild-type strain to high levels of camptothecin but it did not rescue the camptothecin hypersensitivity of a *rad51Δ* strain (Figs 4C and EV2A and B). These data were thus consistent with our conclusions that camptothecin sensitivity is not mainly generated via the rDNA or telomeric loci. Moreover, they indicated that the features of the chromatin template that are toxic to *tof1Δ* and wild-type cells in the presence of camptothecin are generated in a Sir1-dependent manner.

## Various SIR-bound genomic regions mediate camptothecin sensitivity

Our findings suggested that loss of the SIR complex might promote camptothecin resistance via effects on the *HM* loci. To test whether this might be connected to changes on the *HM* chromatin template itself or associated expression of genetic information from the normally silenced *HML* locus, we analysed the sensitivity of diploid *tof1Δ/tof1Δ* cells that simultaneously express the genetic information encoded by *HMR* and *HML*. If transcription of genetic information from the *HML* locus reduced the camptothecin hypersensitivity of MATa *tof1Δ* strains, one would expect a homozygous *tof1Δ* diploid strain to be less camptothecin sensitive than the corresponding haploid strain; however, this was not the case (Fig 4E). Moreover, the camptothecin hypersensitivity of diploid *tof1Δ/tof1Δ* cells was also rescued by sirtinol, clearly establishing that chromatin alterations, rather than expression of *HM* genetic information, are responsible for suppression of camptothecin hypersensitivity (Fig 4E).

While the above findings suggested that the chromatin status of the *HM* locus governs camptothecin sensitivity, when we deleted the *HML* and *HMR* loci, we were surprised to observe that this did not rescue the camptothecin hypersensitivity of *tof1Δ* cells (Fig 5A). This observation therefore strongly suggested the existence of other genomic loci targeted by Sir1-4 as governing camptothecin sensitivity. To attempt to identify such loci, we analysed chromatin immunoprecipitation-sequencing (ChIP-seq) data for Sir2, Sir3, Sir4, GFP, acetylated histone H4-K16, and histone H3 [38,39]. In these datasets, we searched for genomic regions displaying higher association with Sir2, Sir3, and Sir4 compared to neighbouring regions. From the ensuing list, we then removed regions displaying increased GFP binding to exclude ChIP bias towards highly expressed genes [39]. We also removed regions where we did not observe decreased histone H4-K16 acetylation (the consequence SIR complex binding) compared to neighbouring regions, as well as regions also displaying reduced histone H3 ChIP signals suggesting depletion of nucleosomes. Genomic regions identified in this manner localised to confirmed open reading frames (ORFs; Figs 5B and EV2C).

We then defined a "SIR-binding score" (the fraction of nucleotides for which the above conditions held) for every ORF in the yeast genome. While the majority of all ORFs essentially had a null SIR score (indicative of no enrichment of SIR complex binding), we found that 82 of them showed an enrichment of Sir2/3/4 and concomitant loss of H4-K16 acetylation along more than 20% of their sequence (Table EV4). Of these 82 ORFs, 28 were localised in subtelomeric regions or in regions proximal to the *HM* loci (Fig 5C, small grey dots), while the remaining 54 hits were positioned along chromosome lengths (Fig 5C, green dots). Although the majority of the identified ORFs are expressed at high levels during exponential growth, high expression was not sufficient for a high SIR score (Fig EV2D based on data from [40]). Taken together, these findings highlighted how, in addition to functioning at its well-defined target loci, the SIR complex may also act at a variety of loci scattered throughout the genome, and suggested that these loci might also promote camptothecin toxicity in wild-type and *tof1Δ* cells.

Recruitment of Sir1 at *HM* loci requires its interaction with the bromo-adjacent domain (BAH) region of Orc1 [36,41,42]. We therefore assessed whether any of the loci we identified above were also positioned in proximity to a site bound by ORC. Thus, we calculated the distance between the centre of each ORF and the nearest ORC binding site [43]. This revealed that ~50% of SIR-enriched ORFs were located less than ~750 bp from a site of ORC binding (Fig 5D), a distance considerably shorter than the median value of 7.7 kbp for all yeast ORFs. We therefore reasoned that, if ORC has a functional role in recruiting the SIR complex to these genomic loci, it should be possible to suppress the camptothecin hypersensitivity of *tof1Δ* cells by preventing ORC-mediated recruitment of Sir1. In line with this hypothesis—and in contrast to what we had observed upon deleting the HM locus—deleting the BAH domain of Orc1 markedly suppressed the camptothecin hypersensitivity of *tof1Δ* cells (Fig 5E; effects of *ORC1* deletion could not be studied because it is an essential gene). In agreement with the proposed role of Orc1 BAH domain in recruiting the SIR complex, deletion of *SIR2* did not further enhance camptothecin resistance of *orc1ΔBAH* cells, either in a wild-type or a *tof1Δ* background (Fig 5F and G). These findings thus suggested that the chromatin substrates that become toxic to *tof1Δ*

cells exposed to camptothecin are at least partially formed in an ORC-dependent manner.

## Evidence for an evolutionarily conserved connection between sirtuin function and camptothecin sensitivity

Consistent with a model in which SIR proteins might promote camptothecin sensitivity in other organisms, we found that *sir2Δ* *S. pombe* strains were more resistant to camptothecin than control *sir2*[+] strains (Fig 6A). Moreover, in line with our findings in budding yeast, addition of sirtinol to the growth medium reduced the camptothecin sensitivity of wild-type *S. pombe* cells (Fig 6B).

Next, we tested whether sirtinol affected the camptothecin sensitivity of hTERT-immortalised, non-transformed human RPE-1 cells. In line with previous studies [44,45], we found that sirtinol induced a dose-dependent killing of human cells (Fig 6C). However, when the cells were pre-incubated with sirtinol for 24 h prior to camptothecin addition, a protective effect on the survival to camptothecin was observed (Fig 6D). As expected, flow cytometry analysis based on DNA content showed that camptothecin induced a replication-dependent cell cycle arrest, with the majority of cells in G2/M after 24 h of camptothecin treatment. By contrast, camptothecin-induced G2/M accumulation was much less pronounced when cells had been pre-treated with sirtinol (Fig 6E). Importantly, when we used EdU pulse-labelling to quantify DNA replication, we found that neither the proportion of EdU-incorporating cells nor the average intensity of EdU per cell was significantly affected by sirtinol (Fig 6F), indicating that replication was not inhibited by the sirtinol concentration used in our experimental setting. Collectively, these results supported a model in which cell killing induced by camptothecin is to a large degree mediated by the action of sirtuins, via a mechanism that is conserved from yeast to human cells.

## Discussion

By using a synthetic viability screening approach [21], we identified the *SIR3* and *SIR4* genes as major mediators of the sensitivity of both wild-type and *tof1Δ* cells to camptothecin. We subsequently established that *SIR2* and *SIR1* also function in a similar way (these genes were likely not found in our initial screen because of the relatively small number of suppressor strains analysed). We established that, rather than by reducing camptothecin action, deletion of these *SIR* genes removes a factor that hinders cell proliferation in the presence of camptothecin in wild-type cells and that is particularly toxic to cells lacking the Tof1-Csm3 replication-pausing complex. Camptothecin promotes the accumulation of positive supercoiling during DNA replication by locking topoisomerase 1 on DNA in a non-functional state [19,20]. Since Tof1 and Csm3 function to restrict replisome rotation during DNA replication [9], and since an important factor driving fork rotation is positive supercoiling [19], we hypothesise that an excess of camptothecin-induced positive supercoiling is the factor that is alleviated by deletion of *SIR* genes.

While camptothecin is a Top1 inhibitor, lack of Top1 activity is not sufficient to explain the hypersensitivity of *tof1Δ* and *csm3Δ* cells, as deletion of *TOP1* is not toxic to *tof1Δ* cells. Instead, camptothecin-mediated locking of Top1 on DNA could be the source of topological stress, either directly by creating topologically closed

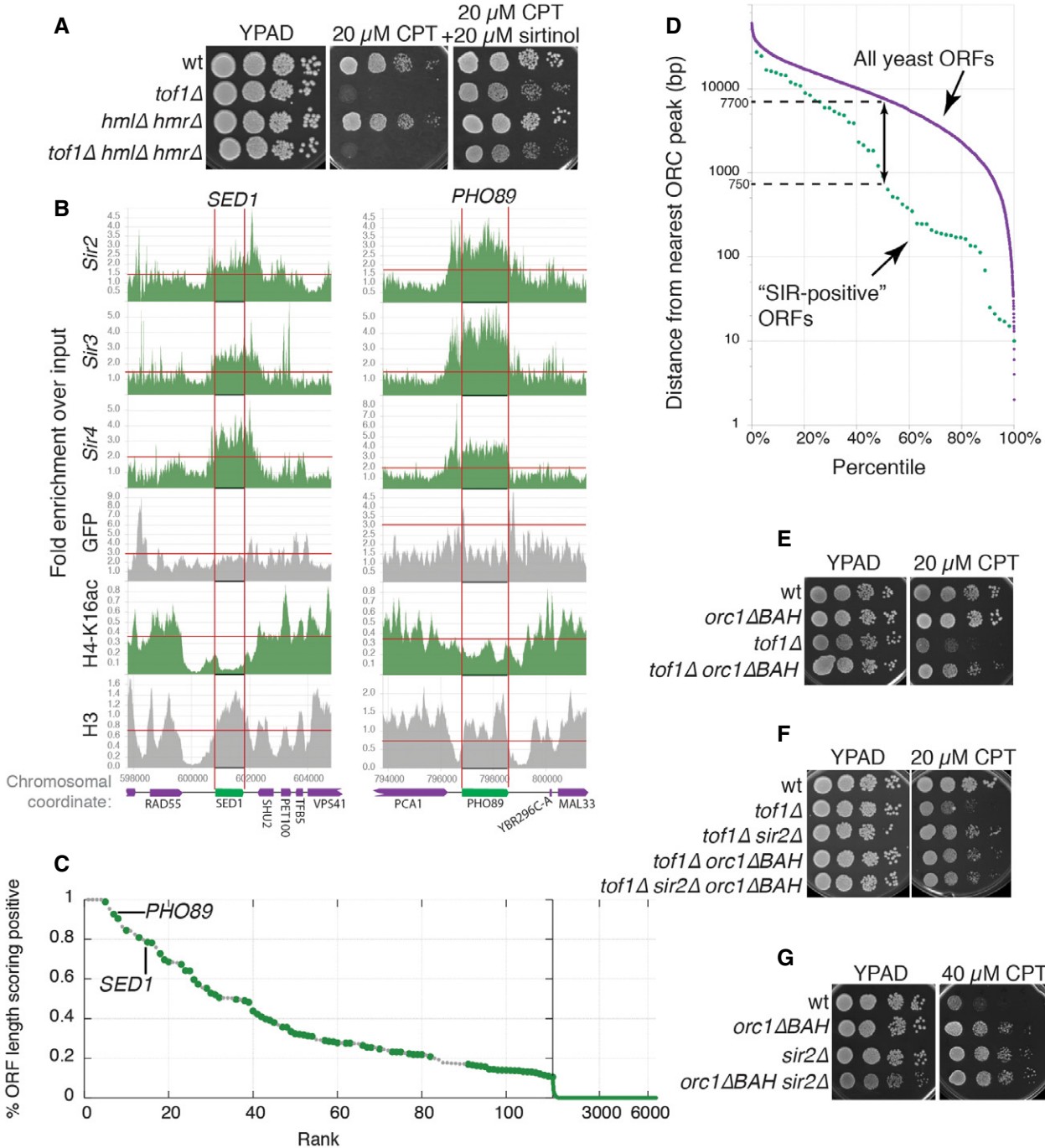

**Figure 5.  Disruption of ORC1-mediated binding of the SIR complex to highly transcribed genes suppresses camptothecin hypersensitivity of *tof1Δ* cells.**

A   Deletion of *HML* and *HMR* does not suppress camptothecin hypersensitivity of *tof1Δ* strains.

B   Analysis of ChIP-seq data for the proteins indicated on the *y*-axes. Enrichments are plotted as a function of the genomic coordinate; in green is the protein/modification tested; in grey are controls. The ORFs indicated at the top of each graph are identified by two vertical red lines. The horizontal red lines indicate the thresholds used in this work to determine enrichment (Sir2/Sir3/Sir4/GFP/H3) or loss (H4-K16) of ChIP signal.

C   Identification of regions bound by the SIR complex: for each ORF in the genome, a "SIR score" was calculated as the fraction of the ORF for which both increased Sir2, Sir3, Sir4, and decreased H4-K16ac were observed. ORFs were sorted based of their "SIR score". Subtelomeric ORFs and ORFs proximal to *HML* and *HMR* are shown with small grey dots, while remaining ORFs are shown with large green dots.

D   SIR-positive ORFs are on average located closer to sites of ORC binding than ORFs in general. All yeast ORFs are shown in purple as a function of their distance from the nearest site of ORC binding. Non-telomeric and non-HM SIR-positive ORFs (SIR score > 0.2) are shown in green.

E   Deletion of the BAH domain of ORC1 partially rescues camptothecin hypersensitivity of *tof1Δ* cells.

F   In a *tof1Δorc1ΔBAH* background, deletion of *SIR2* does not further increase camptothecin resistance.

G   *SIR2* deletion and *orc1ΔBAH* mutation are epistatic with regard to camptothecin resistance.

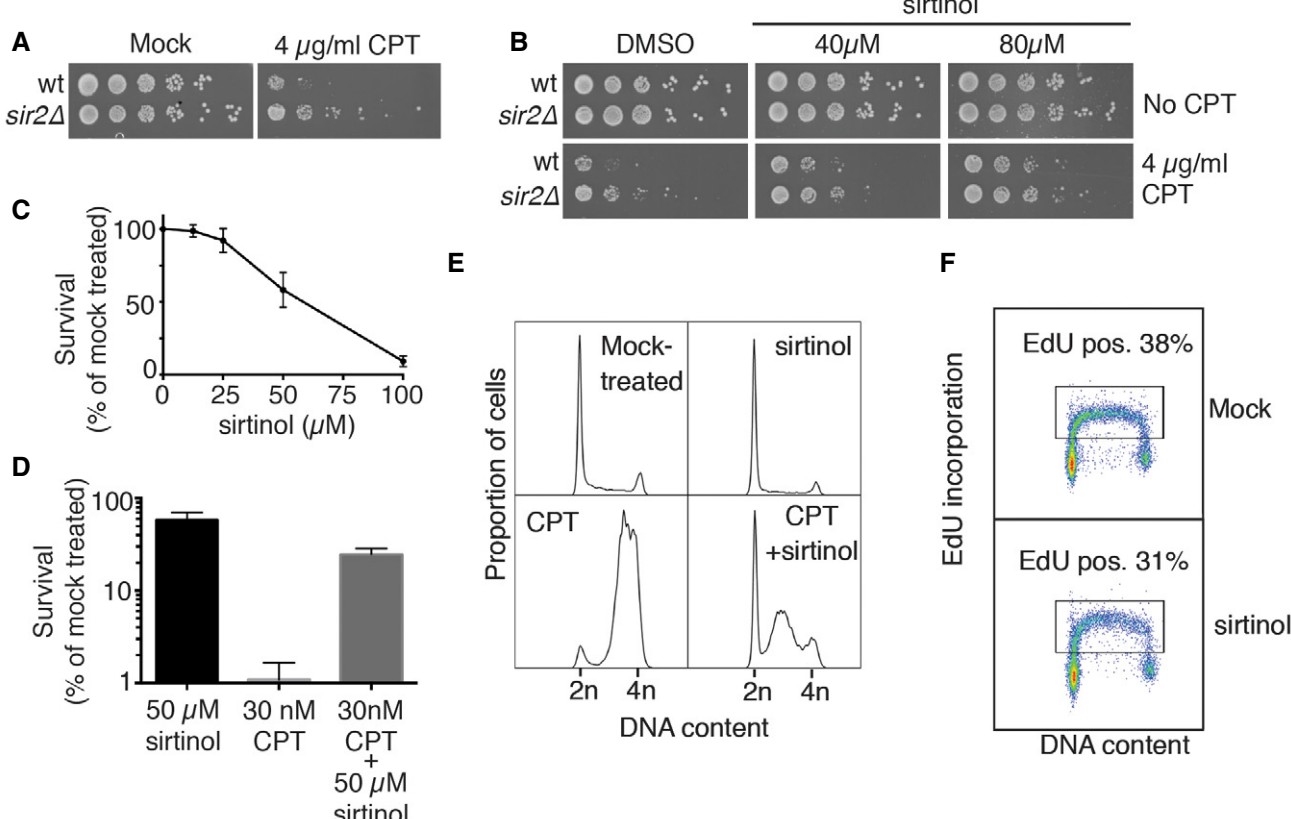

**Figure 6. The role of sirtuins in driving camptothecin sensitivity appears to be evolutionarily conserved.**

A  *Schizosaccharomyces pombe sir2Δ* cells are more resistant to camptothecin than congenic wild-type controls.

B  Sirtinol alleviates camptothecin sensitivity of wild-type *S. pombe* cells.

C  Sirtinol is cytotoxic for non-transformed human cells. Clonogenic capacity of cells was measured after 48-h treatment with indicated doses of sirtinol. Average and standard deviation ($n = 4$) are shown for each point.

D  Sirtinol rescues camptothecin-mediated lethality in human cells. Cells were pre-treated with 50 μM sirtinol for 24 h and then incubated for another 24 h in the presence of 50 μM sirtinol, 30 nM camptothecin or a combination of both drugs. Average and standard deviation ($n = 4$) are shown for each point.

E  48-h treatment with sirtinol does not affect the cell cycle distribution, but partially alleviates the G2/M arrest caused by camptothecin. A representative experiment is shown.

F  Sirtinol does not inhibit DNA replication in hTERT RPE-1 cells. A representative experiment is shown.

domains, or indirectly by preventing Top2 action—similar to what is thought to happen when a catalytically inactive Top2 that still retains an ability to bind DNA is expressed [46].

Lack of Sir2, Sir3, or Sir4 leads to loss of histone H4 lysine 16 (H4-K16) deacetylation and subsequent impairment in heterochromatin formation. We have observed that inhibition of Sir2 deacetylase activity or mutation of H4-K16 to glutamine (a residue that mimics an acetylated lysine) also increases the camptothecin resistance of both wild-type and *tof1Δ* cells. Importantly, we have established that the cell cycle delay observed in camptothecin-treated *tof1Δ* cells can only be suppressed if Sir2 activity is inhibited prior to camptothecin treatment, suggesting that it is the state of the chromatin template itself that becomes toxic to cells when replicated in the presence of camptothecin.

Yeast genomes contain three well-characterised regions of transcriptionally silenced chromatin: the ribosomal DNA, subtelomeric regions and the cryptic mating-type loci *HML* and *HMR*; and of these, only the cryptic mating-type loci require Sir1 for their

silencing [37]. The fact that *SIR1* deletion also suppresses the camptothecin sensitivity of *tof1Δ* cells initially suggested to us that *HML* and *HMR* represent the chromatin templates that are toxic to *tof1Δ* cells in the presence of camptothecin. However, we did not observe a reduction in *tof1Δ* sensitivity to camptothecin by deleting *HML* and *HMR*, meaning that these two genomic loci alone are not responsible for the strong camptothecin sensitivity phenotype displayed by *tof1Δ* cells.

Analysis of publicly available ChIP-seq data allowed us to identify various genomic loci that exhibit enhanced localisation of Sir2, Sir3 and Sir4 as well as H4-K16 under-acetylation. Notably, we found that these genomic loci colocalise with confirmed ORFs and are located closer to sites of ORC binding than the average yeast ORF. Indeed, we found that many of these sites colocalise with genomic loci that were previously shown to bind ORC despite not having replication origin activity [43]. Importantly, we note that some of the SIR-enriched loci also colocalise with sites of replication fork pausing and sites enriched in binding of Rrm3, a DNA helicase

that relieves replication fork pauses [47,48], suggesting that SIR-enriched loci are inherently difficult to replicate even in the absence of camptothecin. The fact that these ORFs are amongst the most highly expressed yeast genes and yet exhibit enhanced recruitment of the SIR silencing complex and markers of histone H4 deacetylation is enigmatic. One possibility is that strong transcription could prevent heterochromatin formation despite the presence of the SIR complex. Indeed, it has been shown that promoter strength affects the efficiency of silencing [49]. In this regard, the sensitivity of yeast cells to camptothecin might stem from DNA catenation that is generated when replication forks approach barriers created by the Sir2/3/4 complex, a phenotype that would be exacerbated by the absence of the Tof1/Csm3 complex. We note that increased catenation would likely require time to be resolved, thereby potentially accounting for the M/G1 delay observed following camptothecin treatment in wild-type cells and more strongly in *tof1Δ* cells. Another possibility is that SIR-mediated genomic loci could be particularly prone to replication-induced topological stress and would therefore be more frequently targeted by Top1 and more susceptible to camptothecin-induced DNA damage. In this context, replisome instability caused by *TOF1* deletion would increase the chance of fork breakdown or failure to rescue fork reversal events.

We have also provided evidence that the role of sirtuins in driving camptothecin sensitivity is evolutionarily conserved from yeast to humans. In the case of *S. pombe*, either loss or inhibition of Sir2, the fission yeast *SIR2* ortholog, results in camptothecin resistance. As in *S. cerevisiae*, fission yeast Sir2 is involved in the heterochromatin assembly within the mating-type locus, subtelomeric regions and centromeric DNA [50,51] by deacetylating histone H3-K9 and histone H4-K16. Furthermore, we have found that sirtinol protects non-transformed human RPE-1 cells from killing by camptothecin via a mechanism that does not appear to reflect effects on DNA replication per se. As in yeast, human SIRT1 and SIRT2 deacetylate H3-K9 and H4-K16, among other substrates, and promote heterochromatin formation and gene silencing [52,53]. This suggests that the role of sirtuins in camptothecin-mediated lethality in human cells may be similar to that in yeast, although alternative mechanisms cannot be ruled out due to the considerable number of sirtuin-dependent pathways documented in mammalian systems.

Inhibition of topoisomerase 1 is a widely used therapeutic strategy to selectively kill proliferating cancer cells, with camptothecin analogues being part of the standard of care provided by many cancer clinics worldwide. Various mechanisms of camptothecin resistance have been observed, ranging from overexpression of drug-efflux transporters, which actively reduce intracellular drug concentration [54], to specific Top1 mutations that prevent its interaction with camptothecin [55,56]. On the other hand, several sirtuin inhibitors have been shown to exhibit cytotoxic activity against various cancer cell lines (reviewed in [57]) and are currently being assessed for their potential clinical applicability [58,59]. Using budding yeast as a model system, we have found that inhibition of histone H4-K16 deacetylation by inactivation of the SIR protein complex represents an additional mechanism of camptothecin resistance and that this mechanism is likely conserved in fission yeast and in human cells. Further studies will be required to determine the precise mechanism-of-action of sirtinol in both transformed and non-transformed human cells and whether sirtuins play a role in the emergence of resistance to camptothecin analogues in cancers.

# Materials and Methods

### Yeast strains and plasmids

Yeast strains used for this work are haploid derivatives of W303 unless otherwise indicated and are listed in Table EV1. All deletions were introduced by one-step gene disruption/tagging [60]. Strains carrying histone H4 mutations were obtained by plasmid shuffling, transforming the strain JHY6 (*hht1-hhf1Δ::KanMX6 hht2-hhf2Δ:: HPH*) with plasmids obtained by site-directed mutagenesis of plasmid pMR206 (*HHT2-HHF2; TRP1*). Isolation of suppressor strains was carried out as previously described [21]. The number of colonies sequenced was determined by reason of economics. *S. pombe* strains used were 49 (*h+ ade6-M210 leu1-32 ura4-D18*) and 34 (*h+ sir2::kanMX6 ade6-M216 leu1-32 ura4-D18*).

### Whole-genome paired-end DNA sequencing and data analysis

Whole-genome paired-end DNA sequencing and data analysis were performed as previously described [21]. All raw sequencing data are available from the European Nucleotide Archive (ENA) under the accession codes detailed in Table EV2. SNPs and indels were identified by using the SAMtools (v0.1.19) mpileup function, which finds putative variants and indels from alignments and assigns likelihoods, and BCFtools that performs the variant calling [61]. The following parameters were used: for SAMtools (v0.1.19) mpileup "-EDS -C50 -m2 -F0.0005 -d 10000" and for BCFtools (v0.1.19) view "-p 0.99 –vcgN". Functional consequences of the variants were produced by using the Ensembl VEP [62].

### Drug sensitivity assays

Overnight-grown saturated cultures of the indicated strains were serially diluted (10 fold) in water; 10-μl drops of each dilution were deposited on each plate. Images were scanned 2–3 days after plating and growth at 30°C. Each experiment was repeated at least twice ($n \geq 2$).

### Analysis of yeast cell cycle progression and Western blotting

Exponentially growing cultures (30°C) were synchronised in G1 by addition of 5 μg/ml alpha-factor for 2 h. G1-synchronised cultures were then transferred to fresh YPD and released into S phase in the presence or in the absence of camptothecin and/or sirtinol; 45 min after the release, 20 μg/ml alpha-factor was added to allow quantification of G1 cells by preventing re-entry into the cell cycle. To detect Clb2, trichloroacetic acid protein extracts were separated on 10% polyacrylamide gels and Clb2 detection was carried out using anti-Clb2 antibodies (Santa Cruz sc9071).

### Analysis of ChIP-seq data

Reads were aligned using BWA-MEM, and duplicates marked with Picard. For each genomic coordinate, coverage was calculated using samtools and bedtools (samtools view -q10 -b $filename| genomeCoverageBed -d -ibam stdin -g) and normalised using the genomewide median of each sample. For each coordinate, the enrichment (E) was calculated as the ratio of the normalised

coverages of IP and input samples. Every genomic position showing $E_{sir2} > 1.75$ and $E_{sir3} > 1.5$ and $E_{sir4} > 2$ and $E_{GFP} < 3$ and $E_{H4-K16ac} < 0.3$ and $E_{H3} > 0.75$ was exported to a bed file. These values were determined empirically, and small adjustments did not substantially alter the final results. For every ORF, the total number ($T$) of positions (nucleotides) for which the above conditions held was calculated by querying the bed file. The final SIR score was obtained by dividing this number ($T$) by the length of the ORF.

### Human cell culture

hTERT RPE-1 cells were cultured in Dulbecco's modified Eagle's (DME)/F12 1:1 medium (Sigma-Aldrich) supplemented with 10% foetal bovine serum (BioSera), 2 mM L-glutamine, 100 units/ml penicillin and 100 μg/ml streptomycin (Sigma-Aldrich) and buffered with 0.2% $Na(CO_3)_2$.

### Clonogenic survival assays

Cells were treated for 48 h with 50 μM sirtinol (Tocris), with camptothecin 30 nM for 24 h or pre-treated with sirtinol for 24 h and then incubated with camptothecin and sirtinol for another 24 h as indicated. Cells were then washed three times with PBS and left to form colonies for 7–14 days. Colonies were stained with 0.1% (w/v) crystal violet in 20% (v/v) ethanol for counting. Results were normalised to plating efficiencies of untreated cells.

### Human flow cytometry assays

Flow cytometry assays were performed as described in [63]. A 1-h pulse of 10 μM EdU was performed after 48 h of the indicated treatment. After fixation and permeabilisation, Alexa Fluor 488 azide (Invitrogen) was used for the click reaction to detect the incorporated EdU. Finally, cells were resuspended in FACS buffer with DAPI and analysed in a BD LSRFortessa™ cell analyser.

### Data availability: referenced data

ChIP-seq data were downloaded from the Sequence Read Archive (NCBI) using accession numbers specified in Table EV3 and originated from the following publications: (i) Thurtle and Rine [38]; (ii) Teytelman *et al* [39].

**Expanded View** for this article is available online.

### Acknowledgements
We thank Jasper Rine, Catherine A. Fox, Takehiko Kobayashi, and Tony Kouzarides for the gift of strains and plasmids, all the members of the SPJ laboratory for helpful discussions, and Helena Santos-Rosa for advice with designing histone point mutations. Research in the Jackson laboratory is funded by Cancer Research UK [C6/A11224, C6/A18796, C6946/A14492] and the Wellcome Trust [WT092096]; FP was supported by the Associazione Italiana per la Ricerca sul Cancro [10932] and the European Molecular Biology Organization [ALTF1287-2011]; IS, MH, and NJG were supported by the Wellcome Trust [101126/Z/13/Z (Strategic Award—COMSIG)] and [098051 to MH]; VC is supported by the Armenise-Harvard Foundation and the Fondazione Telethon.

### Author contributions
The initial project was conceived by FP, VC and SPJ. Screening for suppressors and DNA extractions were carried out by FP and NJG. Analysis of whole-genome sequencing data was carried out by MH and FP. Subsequent *in vivo* experiments and analyses of ChIP-seq data were designed and carried out by FP; experiments in *S. pombe* and human cells were carried out by IS. The manuscript was largely written by FP, IS and SPJ, with contributions made by all the other authors.

### Conflict of interest
The authors declare that they have no conflict of interest.

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
