## [Review Process File · EMBO Reports]

Manuscript EMBO-2016-43560

Chromatin determinants impart camptothecin sensitivity

Fabio Puddu, Israel Salguero, Mareike Herzog, Nicola J. Geisler, Vincenzo Costanzo, and Stephen P. Jackson

Corresponding authors: Stephen Jackson and Fabio Puddu, The Gurdon Institute and Department of Biochemistry, University of Cambridge

Review timeline:

Submission date:	24 October 2016
Editorial Decision:	29 November 2016
Revision received:	15 February 2017
Editorial Decision:	01 March 2017
Revision received:	07 March 2017
Accepted:	09 March 2017

Editor: Esther Schnapp

Transaction Report:

1st Editorial Decision

29 November 2016

Thank you for the submission of your manuscript to EMBO reports. I am sorry for the slight delay in getting back to you; we have only now received the full set of referee reports, as well as cross-comments.

As you will see from the reports below, while referee 2 is more critical, referees 1 and 3 support publication of your study with only minor revisions.

Upon cross-commenting on each others' reports, referees 1 and 3 also agree that referee 2 is asking for experiments outside the scope of this study, and several of referee 2's concerns therefore do not need to be addressed experimentally. Please refer to the cross-comments by referee 3 (also below) for the concerns of referee 2 that do and do not need to be addressed. Please also address all comments by referees 1 and 3.

Given these constructive comments, we would thus like to invite you to revise your manuscript with the understanding that the referee concerns must be fully addressed (as outlined above) and their suggestions taken on board. Please address all referee concerns in a complete point-by-point response. Acceptance of the manuscript will depend on a positive outcome of a second round of review. It is EMBO reports policy to allow a single round of revision only and acceptance or rejection of the manuscript will therefore depend on the completeness of your responses included in the next, final version of the manuscript.

REFeree REPORTS

Referee #1:

In this manuscript Puddu and colleagues investigate the basis of the camptothecin (CPT) sensitivity of budding yeast cells lacking TOF1 or CSM3. TOF1/CSM3 are the budding yeast orthologues of timeless and tipin in mammals. In all eukaryotic systems tested, mutation of these factors results in replication stress.

The approach used is similar to one used previously to isolate genetic suppressors of Sae2 (1). A genome wide screen for suppressors of tof1/csm3 CPT sensitivity identified loss of function mutation in both SIR3 and SIR4. Subsequent analysis showed that deletion of SIR2 and SIR1 also suppresses the CPT sensitivity of these cells. Therefore, a functional SIR histone deacetylase complex is required for acute sensitivity of tof1/csm3 cells.

To confirm that loss of histone deacetylase activity was the cause of suppression, the authors also mutated the histone residue targeted by the SIR deacetylase, H4-K16. Mutation of this residue also suppressed camptothecin sensitivity. This confirms direct link to chromatin structure. Furthermore, that treatment with sirtinol, which inhibits the enzymatic function of sirtuin complexes, also suppresses CPT sensitivity. The authors then present evidence that a similar mechanism of camptothecin sensitivity exists in fission yeast and human cells.

The study is comprehensive and the experiments well done. The main conclusion that SIR mediated chromatin structures makes cells sensitive to CPT is clearly shown. This finding is important for anybody interested in understanding the pathway of CPT sensitivity and the pathways by which replication stress is generated in cells.

For mechanism, the authors propose that SIR dependent chromatin structures make cell sensitive to topological stress during DNA replication. Top1 has recently been shown to be a key regulator of the response to topological stress in yeast (2). They argue that structures that generate topological stress are more likely to require Top1 action and therefore vulnerable to the inhibition of Top1 that occurs following CPT treatment. There are some issues with this argument, which I will address below. However, I think they are easily dealt with. With minor revision, I recommend this manuscript for publication

Issues:

1) The problem with the argument that CPT inhibits the topological relaxation activity of Top1 is that loss of function mutations in top1 and tof1 do not show the same synthetic interaction. For the topological argument to be correct, there must be some aspect of CPT action on Top1 that increases topological stress beyond that generated by loss of Top1 action alone. Therefore the authors should provide some form of plausible model for why CPT treatment will generate higher levels lethal stress than loss of Top1 alone.

One model would be that the protein-DNA structure generated by CPT treatment at sites of topological stress actually prevents the action of Top2 at these sites ahead of the fork (see issue 2 below). In this way CPT action of Top1 generates higher levels of topological stress than Top1 deletion alone. A similar occlusion of alternate topoisomerase action is proposed to lead to the phenotype of shown when an inactive but DNA binding of Top2 is expressed in cells already lacking Top2 activity (3). The full reasoning behind this argument is provided here (4).

Another plausible model can be based on Top1's proposed role as a replisome stability factor. Here again you would posit that again SIR mediated structures are vulnerable to replication induced topological stress and therefore likely to be frequently targeted by Top1. It has been shown that CPT treatment of yeast cells leads to elevated levels of fork reversal at potentially topologically stressed loci (5). Potentially, increasing fork reversal events in the context of higher replisome instability (from tof1 deletion) could lead to higher levels of fork breakdown or failure to reverse a fork reversal event.

2) In the introduction the authors give the impression that Top1 alone deals with topological stress ahead of the fork whereas Top2 deals with intertwinings generated by fork rotation behind the fork. Although a convenient assumption there is little evidence to indicate that this is true. Top2 in yeast and other organisms is highly proficient at dealing with positive supercoiling both in vitro and in vivo (for example (6)). The mode of action of Top1 suggests it would be more proficient than Top2 at dealing with topological stress ahead of an elongating fork, but Top2 is capable of relaxing this

stress.

A more accurate statement would be "In *Saccharomyces cerevisiae*, the main DNA topoisomerase that relaxes positive supercoiling during DNA replication is considered to be Topoisomerase I (Top1), a type-IB topoisomerase. lacking Top1 can fully replicate their genome because positive supercoils can either be potentially relaxed directly by Top2 supercoil relaxation or by rotating of the fork relative to the DNA, converting impending positive supercoiling into intertwinings/catenation between the two daughter DNA strands

3) In figure 5B the panel is labeled with 80mM sirtinol. Given the other concentrations of sirtinol are in the μM range, I assume this is a typo.

References:

1. F. Puddu et al., Synthetic viability genomic screening defines Sae2 function in DNA repair. *The EMBO Journal*. 34, 1509-1522 (2015).
2. S. A. Schalbetter, S. Mansoubi, A. L. Chambers, J. A. Downs, J. Baxter, Fork rotation and DNA precatenation are restricted during DNA replication to prevent chromosomal instability. *Proc. Natl. Acad. Sci. U.S.A.*, 201505356-15 (2015).
3. J. Baxter, J. F. Diffley, Topoisomerase II inactivation prevents the completion of DNA replication in budding yeast. *Molecular Cell*. 30, 790-802 (2008).
4. J. Baxter, "Breaking up is hard to do": the formation and resolution of sister chromatid intertwinings. *Journal of Molecular Biology*. 427, 590-607 (2015).
5. A. R. Chaudhuri et al., Topoisomerase I: poisoning results in PARP-mediated replication fork reversal. *Nat Struct Mol Biol*. 19, 417-423 (2012).
6. S. L. French et al. Distinguishing the Roles of Topoisomerases I and II in Relief of Transcription-Induced Torsional Stress in Yeast rRNA Genes. *Mol Cell Bio*, 31, 482-494 (2011).

Referee #2:

The authors describe a potentially interesting observation, namely that the Sir complex is partially responsible for yeast and mammalian cell sensitivity to the Top1-blocking agent camptothecin. Due to the fact that sir mutations compensate for CPT sensitivity in *tof1* / *csm3* deletion backgrounds in yeast, the authors propose a connection between replication fork progression, topological stress and heterochromatic domains.

However, the argumentation is mostly based on a limited epistatic analysis of sir mutants and silent domains in yeast, and fails to suggest a mechanistic model underlying the genetic interactions or eliminate an important alternative explanation based on a well-characterized MAT locus heterozygosity phenomenon in budding yeast.

Indeed, it is well established that deletion of Sir complex components leads to HML/HMR expression, which converts haploid budding yeast cells to non-mating MAT α /alpha cells. This is accompanied by a change in DNA DSB repair pathway choice from NHEJ towards HR [\approx str \hat{m} , 1999, *Nature*; Lee, 1999, *Curr Biol*] largely due to down-regulation of a gene involved in NHEJ, NEJ1 [Valencia, 2001, *Nature*]. It is of paramount importance that the authors extensively explore whether sir-dependent HML/HMR de-silencing is responsible for the observed decrease of CPT-sensitivity in sir and *tof1* / *csm3* sir cells. This can be done by introducing a *hml* and *hmr* into haploid WT and sir strains and treating them with CPT. Another essential control will be assessing CPT sensitivity in *nej1* (or other NHEJ gene deletions) in WT or *tof1* / *csm3* backgrounds.

In case the HML/HMR expression is not responsible for the observed phenomena, the authors may also explore a possible effect of Sir on Top1 or other DNA repair protein's acetylation state and stability [akin to Robert T, 2011, *Nature*] as a potential mechanism.

Major comments:

1. The authors suggest that the loss of Sir complex relieves topological stress at certain loci. This claim should be tested by assays measuring DNA topological states.

2. Fig2B: epistasis of sir2 and top1 should be checked under 40mM CPT treatment to show that CPT sensitivity and sir2 -induced rescue is Top1-dependent.
3. Fig2C: the spot assay shows that sir2 -dependent resistance to CPT needs HR (namely Rad51) machinery activity and probably hyper-activates HR. This hints towards a possible mechanism (HR-dependent DSB repair), in the opinion of this reviewer. However, the authors do not comment on this. Moreover, the claim in the text that 'rad51 strain...unable to repair DSBs induced by camptothecin' is not correct: first, there are Rad51-independent HR processes (SSA, BIR) and second, NHEJ or alt-NHEJ may still repair the Top1-bound ends after limited processing by MRX. Therefore, the data on Fig2C and their description should be reconsidered.
4. Fig2D: a spot assay with the next strains should be shown WT, sir2, tof1, tof1 sir2 treated with CPT +/- sirtinol. It is essential to show that sirtinol works through Sir complex in this experimental setup.
5. Fig2(E): Since the Sas2 acetylase is responsible for the H4K16ac modification, the authors should consider its role in the observed phenotypes [Kimura, 2002, Nat Genet; Dang, 2009, Nature]. Moreover rpd3 is known re-establish the silent chromatin in sir2, therefore it also might be used to reinforce the claims of this study [akin to Thurtle-Schmidt, Rine, 2016, Genetics].
6. Fig3D: the data for a WT strain pre-grown with sirtinol and treated as was tof1 should be included.
7. Fig4 and corresponding text: loss of the Sir complex leads to de-silencing (=expression) of the domains that are silent in WT cells (heterochromatic rDNA repeats, telomeres and HML/HMR). Therefore, one may imagine that sir effects might be partially or completely mediated by expression of one or more of these de-silenced domains. Therefore, to address this possibility, the authors should add: to Fig4A - sir2, sir2 rdn1, tof1 sir2, tof1 sir2 rdn1; to the Fig4C: WT, tof1, sir1 and sir1 tof1, and all of these in the hml hmr background. (It is ideal to delete both hmr and hml, or in case of deleting only one of them to make sure that HML contains silent alpha and HMR contains silent a information, which can be checked by mating proficiency, RT-qPCR or sequencing of the loci). Fig4E should have an additional panel of a spot assay with same genotypes as on the present figures but treated with CPT +/-sirtinol.
8. As alternative Sir2-independent roles were proposed for Sir1 [e.g. Sharp, 2003, Genes Dev], the epistasis of sir1 with sir2 in a tof1 background under CPT treatment should be checked. Moreover, sir1 and sir2/3/4 effects in the literature on HR differ significantly, and unexpectedly [Tsukamoto, 1997, Science; Boulton, 1998, EmboJ], which the authors may consider in light of their results.

Referee #3:

In this manuscript, Jackson and colleagues report the unexpected observation that the deacetylation of histone H4K16 determines the sensitivity to the topoisomerase I (Top1) inhibitor camptothecin (CPT) in yeasts and in human cells. Indeed, they show that inactivation of the histone deacetylase Sir2 and other associated SIR proteins increases CPT resistance in both wild type *S. cerevisiae* cells and tof1 mutants, which are unable to prevent fork rotation at CPT-mediated DNA lesions. Remarkably, this phenotype is independent of the role of SIR proteins at the telomeres and at the rDNA array, but it depends on the ORC- and Sir1-dependent recruitment of the SIR complex at >100 ORFs. Moreover, they provide convincing evidence that this role of SIR proteins in CPT sensitivity is conserved in fission yeast and in human cells. Overall, the manuscript is clearly written and the data are convincing. Although the mechanism by which the deacetylation of H4K16 affects fork stability at specific loci remains to be established, this result is important as it is potentially relevant to drug resistance mechanisms in cancer cells treated with CPT and its derivatives. However, important issues need to be addressed prior to publication.

Specific issues:

1. Abstract: the statement that disruption of chromatin domains at silent mating type loci is sufficient to suppress CPT sensitivity is misleading as it is not the case for tof1 mutants (Fig. 4E and p11).

2. Figure 2: How does the CPT sensitivity of *tof1 sir2* cells compare to the sensitivity of *tof1 sir3* or *tof1 sir4*? This is difficult to estimate since these mutants were grown on different plates. How do the authors explain the fact that the *sir2*-mediated suppression is apparently less efficient than the deletion of *Sir3* and *Sir4*?
3. Figure 4H: Is the deletion of the BAH domain of ORC epistatic with the effect of SIR inactivation on CPT sensitivity?
4. P13, line 14: ...to the growth medium reduced...
5. Figure 5A: Are *swi1* and *swi3* mutants HS to CPT in *S. pombe*, as it is the case for *tof1* and *csm3* in human? If so, is this sensitivity suppressed by *sir2* deletion?
6. Page 14, top: Figure 5H should read 5F
7. P15, bottom: ...ChIP-seq data allowed us to identify...

Cross-comments from referee 3:

It seems to me that reviewer #2 is asking for an incredible amount of extra experiments, which do not seem entirely justified to me. For instance his/her point on the potential effect of HML/HMR is valid but was already addressed at least in part in Fig. EV1E. Along the same line, asking for "a possible effect of Sir on Top1 or other DNA repair protein's acetylation state and stability [akin to Robert T, 2011, Nature] as a potential mechanism" is unrealistic. Moreover, the effect reported in Robert 2011 is only observed upon chemical inactivation of several HDACs and not *Sir2* alone. Measuring topological states at multiple chromosomal loci (major comment #1) is also a daunting task that goes way beyond the scope of this study. Same for the effect of *Sas2* and *Rpd3* (major comment #5). Major comment #7 is also partly addressed in the ms. The other points are more relevant and could be performed by the authors.

1st Revision - authors' response

15 February 2017

please find attached our revised manuscript "Chromatin determinants impart camptothecin sensitivity." As you will see, in responding to the reviewers' comments, we have extended and strengthened our manuscript. We therefore hope that you and our reviewers will now feel that our work is suitable for publication as a full article in *EMBO reports*.

REFeree REPORTS

Referee #1:

In this manuscript Puddu and colleagues investigate the basis of the camptothecin (CPT) sensitivity of budding yeast cells lacking *TOF1* or *CSM3*. *TOF1/CSM3* are the budding yeast orthologues of timeless and tipin in mammals. In all eukaryotic systems tested, mutation of these factors results in replication stress. The approach used is similar to one used previously to isolate genetic suppressors of *Sae2* (1). A genome wide screen for suppressors of *tof1/csm3* CPT sensitivity identified loss of function mutation in both *SIR3* and *SIR4*. Subsequent analysis showed that deletion of *SIR2* and *SIR1* also suppresses the CPT sensitivity of these cells. Therefore, a functional SIR histone deacetylase complex is required for acute sensitivity of *tof1/csm3* cells. To confirm that loss of histone deacetylase activity was the cause of suppression, the authors also mutated the histone residue targeted by the SIR deacetylase, H4-K16. Mutation of this residue also suppressed camptothecin sensitivity. This confirms direct link to chromatin structure. Furthermore, that treatment with sirtinol, which inhibits the enzymatic function of sirtuin complexes, also suppresses CPT sensitivity. The authors then present evidence that a similar mechanism of camptothecin sensitivity exists in fission yeast and human cells. The study is comprehensive and the experiments well done. The main conclusion that SIR mediated chromatin structures makes cells sensitive to CPT is clearly shown. This finding is important for anybody interested in understanding the pathway of CPT sensitivity and the pathways by which replication stress is generated in cells. For mechanism, the authors propose that SIR dependent chromatin structures make cell sensitive to topological stress during DNA replication. *Tof1* has recently been shown to be a key regulator of the response to topological stress in yeast (2). They argue that structures that generate topological stress are more likely to require *Top1* action and therefore vulnerable to the inhibition of *Top1* that occurs following

CPT treatment. There are some issues with this argument, which I will address below. However, I think they are easily dealt with. With minor revision, I recommend this manuscript for publication.

We thank this reviewer for his/her positive assessment of our work, and for his/her important and insightful comments and suggestions.

Issues:

1) The problem with the argument that CPT inhibits the topological relaxation activity of Top1 is that loss of function mutations in top1 and tof1 do not show the same synthetic interaction. For the topological argument to be correct, there must be some aspect of CPT action on Top1 that increases topological stress beyond that generated by loss of Top1 action alone. Therefore the authors should provide some form of plausible model for why CPT treatment will generate higher levels of lethal stress than loss of Top1 alone. One model would be that the protein-DNA structure generated by CPT treatment at sites of topological stress actually prevents the action of Top2 at these sites ahead of the fork (see issue 2 below). In this way CPT action of Top1 generates higher levels of topological stress than TOP1 deletion alone. A similar occlusion of alternate topoisomerase action is proposed to lead to the phenotype shown when an inactive but DNA binding Top2 is expressed in cells already lacking Top2 activity (3). The full reasoning behind this argument is provided here (4). Another plausible model can be based on Top1's proposed role as a replisome stability factor. Here again you would posit that again SIR mediated structures are vulnerable to replication induced topological stress and therefore likely to be frequently targeted by Top1. It has been shown that CPT treatment of yeast cells leads to elevated levels of fork reversal at potentially topologically stressed loci (5). Potentially, increasing fork reversal events in the context of higher replisome instability (from tof1 deletion) could lead to higher levels of fork breakdown or failure to reverse a fork reversal event

We thank the reviewer for raising these issues, which, admittedly, we did not discuss adequately in our original manuscript. We agree that there is a fundamental difference between camptothecin-mediated Top1 inhibition and loss of Top1 activity, and that the phenotypes we observe are created by the presence of Top1 covalently bound to DNA rather than mere reduction in Top1 activity. We have included a more detailed discussion of these points on page 15 and 17 of our revised text.

2) In the introduction the authors give the impression that Top1 alone deals with topological stress ahead of the fork whereas Top2 deals with intertwinings generated by fork rotation behind the fork. Although a convenient assumption there is little evidence to indicate that this is true. Top2 in yeast and other organisms is highly proficient at dealing with positive supercoiling both in vitro and in vivo (for example (6)). The mode of action of Top1 suggests it would be more proficient than Top2 at dealing with topological stress ahead of an elongating fork, but Top2 is capable of relaxing this stress

A more accurate statement would be "In *Saccharomyces cerevisiae*, the main DNA topoisomerase that relaxes positive supercoiling during DNA replication is considered to be Topoisomerase 1 (Top1), a type-IB topoisomerase. lacking Top1 can fully replicate their genome because positive supercoils can either be potentially relaxed directly by Top2 supercoil relaxation or by rotating of the fork relative to the DNA, converting impending positive supercoiling into intertwinings/catenation between the two daughter DNA strands.

We apologize for our inaccurate statement in the introduction. As suggested by this reviewer, we have rewritten the corresponding section to include the role of Top2 in dealing with positive supercoils generated by DNA replication.

3) In figure 5B the panel is labeled with 80mM sirtinol. Given the other concentrations of sirtinol are in the μM range, I assume this is a typo.

This was indeed a typo and we have corrected it.

Referee #2:

The authors describe a potentially interesting observation, namely that the Sir complex is partially responsible for yeast and mammalian cell sensitivity to the Top1-blocking agent camptothecin. Due to the fact that sir mutations compensate for CPT sensitivity in *tof1 / csm3* deletion backgrounds in yeast, the authors propose a connection between replication fork progression, topological stress and heterochromatic domains.

However, the argumentation is mostly based on a limited epistatic analysis of sir mutants and silent domains in yeast, and fails to suggest a mechanistic model underlying the genetic interactions or eliminate an important alternative explanation based on a well-characterized MAT locus heterozygosity phenomenon in budding yeast.

Indeed, it is well established that deletion of Sir complex components leads to HML/HMR expression, which converts haploid budding yeast cells to non-mating MAT α /alpha cells. This is accompanied by a change in DNA DSB repair pathway choice from NHEJ towards HR [Strm, 1999, Nature; Lee, 1999, Curr Biol] largely due to down-regulation of a gene involved in NHEJ, NEJ1 [Valencia, 2001, Nature]. It is of paramount importance that the authors extensively explore whether sir-dependent HML/HMR de-silencing is responsible for the observed decrease of CPT-sensitivity in sir and *tof1 / csm3 sir* cells. This can be done by introducing a *hml* and *hmr* into haploid WT and sir strains and treating them with CPT. Another essential control will be assessing CPT sensitivity in *nej1* (or other NHEJ gene deletions) in WT or *tof1 / csm3* backgrounds.

In case the HML/HMR expression is not responsible for the observed phenomena, the authors may also explore a possible effect of Sir on Top1 or other DNA repair protein's acetylation state and stability [akin to Robert T, 2011, Nature] as a potential mechanism.

We thank this reviewer for his/her assessment of our work. We respectfully disagree with Reviewer 2's view that we do not suggest any mechanistic model and that we do not "eliminate an important alternative explanation based on a well-characterized MAT locus heterozygosity."

*The reviewer does make a valid point: SIR2 deletion causes MAT heterozygosity (due to expression of the normally silent HM loci), which might have been able to explain the phenotypic suppression that we observed. Both [Kegel, 2001 Current Biology] and [Valencia, 2001, Nature] showed that the NEJ1 component of the non-homologous end joining machinery is down-regulated in diploid yeast strains when MAT locus expression is heterozygous via loss of the SIR complex. This is due to the binding of the *a1/alpha2* repressor upstream of the NEJ1 gene.*

*However, we clearly show in our manuscript that MAT heterozygosity in a diploid setting does not perceptibly alter the camptothecin hypersensitivity of a *tof1 /tof1* strain. We also show that sirtinol suppresses the camptothecin hypersensitivity of diploid *tof1 /tof1* cells. In such a diploid background, ectopic expression of the HM loci caused by SIR loss does not enhance the formation of the *a1/alpha2* repressor, which is already present in the nucleus.*

We believe the above evidence is sufficient to exclude the possibility that the effects we observed are caused by MAT locus heterozygosity.

Major comments:

1. The authors suggest that the loss of Sir complex relieves topological stress at certain loci. This claim should be tested by assays measuring DNA topological states.

While we think it would be very interesting to quantitatively measure the topological state in different regions of the genome, we feel that such work is beyond the scope of the current manuscript, which is already larger than EMBO Reports guidelines.

2. Fig2B: epistasis of *sir2* and *top1* should be checked under 40mM CPT treatment to show that CPT sensitivity and *sir2*-induced rescue is Top1-dependent.

We have performed the experiment requested by the reviewer. We now show, in Figure EVIE of our revised manuscript, that deletion of SIR2 and deletion of TOP1 suppress camptothecin sensitivity to similar extents, and that the double mutant shows a similar level of suppression to either of the single mutants.

3. Fig2C: the spot assay shows that sir2 -dependent resistance to CPT needs HR (namely Rad51) machinery activity and probably hyper-activates HR. This hints towards a possible mechanism (HR-dependent DSB repair), in the opinion of this reviewer. However, the authors do not comment on this. Moreover, the claim in the text that 'rad51 strain...unable to repair DSBs induced by camptothecin' is not correct: first, there are Rad51-independent HR processes (SSA, BIR) and second, NHEJ or alt-NHEJ may still repair the Top1-bound ends after limited processing by MRX. Therefore, the data on Fig2C and their description should be reconsidered.

From the rad51 experiment this reviewer correctly infers that CPT resistance needs functional homologous recombination (HR) both in the presence or in the absence of Sir2. We did not explicitly comment on this because of the fact that CPT-induced damage requires HR for its resolution is well known and documented. The fact that SIR2 deletion does not change this requirement is in our opinion interesting, and suggests that Sir2 loss does not alter the process(es) that leads from a trapped Top1 to a double stranded DNA break. Indeed, if SIR2 deletion had reduced the amount of DNA damage created, we should have observed an increase in resistance even in the absence of Rad51, similarly to what happens by deleting Top1. We apologize for the misleading wording about rad51 strain being "unable to repair DSB" and we have corrected the description of the figure in the manuscript "Ö a rad51 strain, which is severely defective in repairing DSBs induced by camptothecin".

4. Fig2D: a spot assay with the next strains should be shown WT, sir2 , tof1 , tof1 sir2 treated with CPT +/- sirtinol. It is essential to show that sirtinol works through Sir complex in this experimental setup.

We thank this reviewer for pointing out that Figure 2D was not complete. As suggested, we now show the effect of sirtinol in the presence and absence of camptothecin on wild-type, sir2 , tof1 and tof1 sir2 strains. We show that the suppression mediated by sirtinol seems to be marginally stronger than the suppression mediated by SIR2 deletion. This could suggest that a small fraction of the sirtinol-dependent suppression could arise from the off-target inhibition of other enzymes.

5. Fig2(E): Since the Sas2 acetylase is responsible for the H4K16ac modification, the authors should consider its role in the observed phenotypes [Kimura, 2002, Nat Genet; Dang, 2009, Nature]. Moreover rpd3 is known re-establish the silent chromatin in sir2 , therefore it also might be used to reinforce the claims of this study [akin to Thurtle-Schmidt, Rine, 2016, Genetics].

We agree with this reviewer that it would be very interesting to study the effect of different acetyl-transferases on camptothecin sensitivity, but we believe it will be difficult to obtain incisive results. Sas2 is not the only acetyl-transferase acting on H4K16 (Esa1 is another one for example), and results of over-expression studies that aim to mimic SIR deletion are likely to be influenced by aspecific effects caused by non physiological protein levels. We therefore suggest that the requested analyses be considered beyond the scope of the current study.

6. Fig3D: the data for a WT strain pre-grown with sirtinol and treated as was tof1 should be included.

We did not include this condition in Figure 3 because the purpose of the experiment was to show how sirtinol rescues the defect of a tof1 strain. We now show data for "a wild-type strain pre-grown with sirtinol and treated as was tof1 " in the edited Figure 3. In line with the results from drop assays, this shows that sirtinol alleviates the small amount of mitotic delay induced by camptothecin in a wild-type strain.

7. Fig4 and corresponding text: loss of the Sir complex leads to de-silencing (=expression) of the domains that are silent in WT cells (heterochromatic rDNA repeats, telomeres and HML/HMR). Therefore, one may imagine that sir effects might be partially or completely mediated by expression of one or more of these de-silenced domains. Therefore, to address this possibility, the authors should add: to Fig4A - sir2 , sir2 rdn1 , tof1 sir2 , tof1 sir2 rdn1 ; to the Fig4C: WT, tof1 , sir1 and sir1 tof1 , and all of these in the hml hmr background. (It is ideal to delete both hmr and hml, or in case of deleting only one of them to make sure that HML contains silent alpha and HMR contains silent a information, which can be checked by mating proficiency, RT-qPCR or sequencing of the loci). Fig4E should have an additional panel of a spot assay with same genotypes as on the present

figures but treated with CPT +/-sirtinol.

While it is true that loss of Sir2, Sir3 and Sir4 leads to the de-silencing of several domains, and that loss of silencing from more than one of these domains could be required to observe suppression, as we explain in our manuscript, the results we present and several previous observations argue against this hypothesis. First, rDNA silencing is lost in the absence of Sir2 but not Sir3 or Sir4. Since Sir3 or Sir4 deletion suppresses camptothecin sensitivity, rDNA silencing cannot be responsible for suppression. Second, loss of Sir1 only affects HM silencing (not silencing at telomeres or the rDNA), and loss of Sir1 is sufficient to suppress camptothecin sensitivity. This argues against the loss of telomeric or rDNA silencing as causing this phenotype. As requested, we have added to figure 4E (now renamed as Figure 5A) a panel that shows the effect of sirtinol on the tof1 hml hmr triple mutant.

8. As alternative Sir2-independent roles were proposed for Sir1 [e.g. Sharp, 2003, Genes Dev], the epistasis of sir1 with sir2 in a tof1 background under CPT treatment should be checked. Moreover, sir1 and sir2/3/4 effects in the literature on HR differ significantly, and unexpectedly [Tsukamoto, 1997, Science; Boulton, 1998, EmboJ], which the authors may consider in light of their results.

While it is true that Sir2-independent roles have been proposed for Sir1, our results clearly implicate both Sir1 and Sir2 (and the rest of the Sir complex). Moreover, loss of Sir1 or Sir2 (or Sir3 or Sir4) results in very similar levels of suppression (as we show in Figure EV2B). We therefore think it is extremely unlikely that deletion of two factors functionally interacting with each other would result in very similar effects for different and unrelated reasons. In any case, we show in the revised version of the manuscript (Figure 4D) that SIR1 deletion does not improve the camptothecin survival of a tof1 sir2 strain.

Referee #3:

In this manuscript, Jackson and colleagues report the unexpected observation that the deacetylation of histone H4K16 determines the sensitivity to the topoisomerase I (Top1) inhibitor camptothecin (CPT) in yeasts and in human cells. Indeed, they show that inactivation of the histone deacetylase Sir2 and other associated SIR proteins increases CPT resistance in both wild type *S. cerevisiae* cells and tof1 mutants, which are unable to prevent fork rotation at CPT-mediated DNA lesions.

Remarkably, this phenotype is independent of the role of SIR proteins at the telomeres and at the rDNA array, but it depends on the ORC- and Sir1-dependent recruitment of the SIR complex at >100 ORFs. Moreover, they provide convincing evidence that this role of SIR proteins in CPT sensitivity is conserved in fission yeast and in human cells. Overall, the manuscript is clearly written and the data are convincing. Although the mechanism by which the deacetylation of H4K16 affects fork stability at specific loci remains to be established, this result is important as it is potentially relevant to drug resistance mechanisms in cancer cells treated with CPT and its derivatives. However, important issues need to be addressed prior to publication.

We thank this reviewer for his/her positive assessment of our work.

Specific issues:

1. Abstract: the statement that disruption of chromatin domains at silent mating type loci is sufficient to suppress CPT sensitivity is misleading as it is not the case for tof1 mutants (Fig. 4E and p11).

We apologize for this misleading sentence in the abstract, which we have now removed. We meant to say that disruption of all Sir1-dependent chromatin domains is required to observe the suppression and that these domains are normally established at mating type loci and possibly in other regions of the genome.

2. Figure 2: How does the CPT sensitivity of tof1 sir2 cells compare to the sensitivity of tof1 sir3 or tof1 sir4? This is difficult to estimate since these mutants were grown on different plates. How do the authors explain the fact that the sir2-mediated suppression is apparently less efficient than the deletion of Sir3 and Sir4?

Because of the low water solubility of camptothecin and experiments being carried out at different times with different camptothecin batches, one cannot directly compare the data in different figures (the reviewer was comparing Figures 2A and 2B). For this reason, we have tested the sensitivity tof1 sir2, tof1 sir3, tof1 sir4, and tof1 sir1 on the same plate. As we discuss in our revised text, the ensuing results, shown in supplementary figure EV2B, reveal very similar levels of suppression mediated by the different SIR mutant backgrounds.

3. Figure 4H: Is the deletion of the BAH domain of ORC epistatic with the effect of SIR inactivation on CPT sensitivity?

We thank the reviewer for suggesting this experiment. We now show in our revised manuscript (Figures 5F and G) that deleting SIR2 in either a orc1 BAH and in a tof1 orc1 BAH background does not further enhance camptothecin resistance, in agreement with the established model that Sir1 and the Sir2/3/4 complex act downstream of ORC1.

4. P13, line 14: ...to the growth medium reduced...

We have corrected this typographical error.

5. Figure 5A: Are swi1 and swi3 mutants HS to CPT in *S. pombe*, as it is the case for tof1 and csm3 in human? If so, is this sensitivity suppressed by sir2 deletion?

*We thank the Referee 3 for these questions. We have checked the sensitivity of swi1 and swi3 mutants, and we indeed found that they are hypersensitive to camptothecin; however their hypersensitivity did not seem to be suppressed by sirtinol or sir2 deletion. We find difficult to define the exact reason for this lack of suppression. We noticed that *S. pombe* swi1 and swi3 mutants are considerably more sensitive to camptothecin than their *S. cerevisiae* counterparts, so we think it is possible that *S. pombe* Swi1 and Swi3 may have additional functions which are crucial for effective DSB repair (one example is that *S. pombe* but not *S. cerevisiae* Mrc1 is required for survival in the presence of camptothecin). Alternatively or in addition, we note that compared to *S. cerevisiae*, *S. pombe* has further pathways/factors involved in heterochromatin formation, which could become more important in the absence of the Swi1/Swi3 complex. As it stands, we have not included the new data in our revised manuscript because we feel that they would likely require a lengthy discussion and would not add much in the way of insights/conclusions arising from our work. We will, however, be happy to include them should this reviewer and/or the editor feel it necessary.*

6. Page 14, top: Figure 5H should read 5F

7. P15, bottom: ...ChIP-seq data allowed us to identify...

We have corrected these typographical errors.

Cross-comments from referee 3:

It seems to me that reviewer #2 is asking for an incredible amount of extra experiments, which do not seem entirely justified to me. For instance his/her point on the potential effect of HML/HMR is valid but was already addressed at least in part in Fig. EV1E. Along the same line, asking for "a possible effect of Sir on Top1 or other DNA repair protein's acetylation state and stability [akin to Robert T, 2011, Nature] as a potential mechanism" is unrealistic. Moreover, the effect reported in Robert 2011 is only observed upon chemical inactivation of several HDACs and not Sir2 alone. Measuring topological states at multiple chromosomal loci (major comment #1) is also a daunting task that goes way beyond the scope of this study. Same for the effect of Sas2 and Rpd3 (major comment #5). Major comment #7 is also partly addressed in the ms. The other points are more relevant and could be performed by the authors.

We thank Referee 3 for his/her constructive comments.

Thank you for the submission of your revised manuscript to EMBO reports. We have now received the comments from referee 3 who was asked to assess it. As you will see, s/he only has one more minor comment that should be addressed in the manuscript text before we can proceed with its official acceptance.

Please change all color in the EV tables to black, white and grey color. We can unfortunately not process tables with other colors. Tables EV2 and 3 are strangely formatted. It would be better if they could be fit on one page. Please submit all tables as either word or excel files.

The 3 gel pieces shown in Figure 3B seem to be assembled from different pieces, as the borders of each gel piece are not aligned. Can you please explain what happened and send us the source data for this figure panel?

In Fig 5B the label "chromosomal coordinate" is somehow misplaced. In Figs 5B and EV2C some of the numbers on the y-axis in the right column are a merge of two numbers at once. Can this be corrected?

In Fig 6F the label should probably read "incorporation."

Can you please send us an ORCID ID for yourself? This seems to be missing.

EMBO press papers are accompanied online by A) a short (1-2 sentences) summary of the findings and their significance, B) 2-3 bullet points highlighting key results and C) a synopsis image that is 550x200-400 pixels large (the height is variable). You can either show a model or key data in the synopsis image. Please note that text needs to be readable at the final size. Please send us this information along with the revised manuscript.

When you upload a new version of your manuscript in our online system, you can bring forward the current files automatically, and do not need to upload everything from scratch. The files can then be replaced with new files where necessary. Please let us know if you have any questions.

We have uploaded a new revision of the manuscript. Please find our replies to your questions below. If you notice we missed anything else, please let us know. Many thanks.

POINT-BY-POINT RESPONSE

Thank you for the submission of your revised manuscript to EMBO reports. We have now received the comments from referee 3 who was asked to assess it. As you will see, s/he only has one more minor comment that should be addressed in the manuscript text before we can proceed with its official acceptance.

We have included in the introduction the findings reported in the article from the Hyrien lab that referee 3 suggested (Lucas et al, 2001 EMBO J. 20, 6509).

Please change all color in the EV tables to black, white and grey color. We can unfortunately not process tables with other colors.

Tables EV2 and 3 are strangely formatted. It would be better if they could be fit on one page. Please submit all tables as either word or excel files.

I have removed colors from Tables EV2 and EV3 and I have uploaded excel files for every table. I did not include these before because the automated pdf conversion seems to spread the table across different pages.

The 3 gel pieces shown in Figure 3B seem to be assembled from different pieces, as the borders of each gel piece are not aligned. Can you please explain what happened and send us the source data for this figure panel?

The three gel pieces in Figure 3B come from two different gels run in parallel with samples coming from the same experiment. I uploaded the source data, so that you can get a better idea of what I mean. Please note that in the film uploaded there is also a western blot for H3-K56 acetylation and results for another strain (mrc1). These were not included in the final version of the manuscript.

In Fig 5B the label "chromosomal coordinate" is somehow misplaced.

I have corrected this

Figs 5B and EV2C some of the numbers on the y-axis in the right column are a merge of two numbers at once. Can this be corrected?

I have corrected this

Fig 6F the label should probably read "incorporation."

I have corrected this

EMBO press papers are accompanied online by A) a short (1-2 sentences) summary of the findings and their significance, B) 2-3 bullet points highlighting key results and C) a synopsis image that is 550x200-400 pixels large (the height is variable). You can either show a model or key data in the synopsis image. Please note that text needs to be readable at the final size. Please send us this information along with the revised manuscript.

Please find attached to this email the text and our synopsis image.

When you upload a new version of your manuscript in our online system, you can bring forward the current files automatically, and do not need to upload everything from scratch. The files can then be replaced with new files where necessary. Please let us know if you have any questions. I look forward to seeing a final version of your manuscript as soon as possible.

3rd Editorial Decision

09 March 2017

I am very pleased to accept your manuscript for publication in the next available issue of EMBO reports. Thank you for your contribution to our journal.

YOU MUST COMPLETE ALL CELLS WITH A PINK BACKGROUND

Corresponding Author Name: Stephen P Jackson and Fabio Puddu

Manuscript Number: EMBOR-2016-43560